# Glycation Improved the Interfacial Adsorption and Emulsifying Performance of β-Conglycinin to Stabilize the High Internal Phase Emulsions

**DOI:** 10.3390/foods12142706

**Published:** 2023-07-14

**Authors:** Hongjian Zhang, Yan Tian, Siyi Pan, Lianhe Zheng

**Affiliations:** 1College of Food Science and Engineering, Hainan University, Haikou 570228, China; 21110832000013@hainanu.edu.cn; 2Hainan Institute of Grain and Oil Science, Qionghai 571400, China; 3College of Food Science and Technology, Huazhong Agricultural University, Wuhan 430070, China

**Keywords:** β-conglycinin (7S), glycation, interfacial adsorption, emulsifying performance, high internal phase emulsions

## Abstract

This study investigated the interfacial adsorption and emulsifying performance of glycated β-conglycinin (7S) with D-galactose (Gal) at various times. Results indicated that glycation increased the particle sizes and zeta potentials of glycated 7S by inducing subunit dissociation. Glycation destroyed the tertiary structures and transformed secondary structures from an ordered one to a disordered one, leading to the more flexible structures of glycated 7S compared with untreated 7S. All these results affected the structural unfolding and rearrangement of glycated 7S at the oil/water interface. Therefore, glycated 7S improved interfacial adsorption and formed an interfacial viscoelasticity layer, increasing emulsifying performance to stabilize high internal phase emulsions (HIPE) with self-supportive structures. Furthermore, the solid gel-like network of HIPE stabilized by glycated 7S led to emulsification stability. This result provided new ideas to improve the functional properties of plant proteins by changing the interfacial structure.

## 1. Introduction

High internal phase emulsions (HIPE), characterized by an internal phase volume percentage of more than 74%, have obtained remarkable attention due to many advantages, such as long-term storage stability, delivering bioactive components, environmental friendliness, and promising potentials in the food, pharmaceutical, and cosmetic industries [1]. Protein–polysaccharide complexes or conjugates have been justified as excellent emulsifiers to stabilize the HIPE [2]. Especially, multiple investigations have demonstrated that protein–polysaccharide conjugates exhibited enhanced interfacial adsorption and accumulation and the formation of viscoelastic interfacial layers, remarkably improving the emulsifying capability [3]. For instance, glycated proteins, such as soy protein isolates [4], soy glycinin [5], ovalbumin [6], and pecan protein [7], showed improved emulsifying efficiency for stabilizing HIPE with thermal and antioxidant stability or coalescence stability against storage. However, the stabilization mechanism of HIPE by glycated proteins with enhanced emulsification activity was still unknown.

Glycation treatment can change the physicochemical properties of proteins, such as particle sizes, net charges, and surface hydrophobicity, thus affecting emulsification activity. For example, net charges, surface hydrophobicity, and emulsifying activity were significantly improved after coconut protein isolate glycosylation with pectin [8]. Compared with untreated ovalbumins, glycation with fructose decreased the particle sizes and net charges and increased surface hydrophobicity of the ovalbumins, and, therefore, conjugate-stabilized HIPE exhibited smaller droplet sizes and better freeze–thaw stability [6]. In addition, many studies proved a correlation between protein interfacial properties and emulsification activity, i.e., the higher the interfacial activity, the better the emulsification activity. For instance, glycated pea protein isolates [9] and soy protein isolates [10] exhibited better interfacial adsorption and the ability to form thicker viscoelastic films at the interface, enhancing the emulsifying efficiency as well as stability. Therefore, we proposed a hypothesis that characterizing the interfacial adsorption behavior of glycated proteins at the oil/water interface was beneficial for understanding the reasons for improved emulsifying activity.

Many studies indicated that soybean protein isolates (SPI) had good emulsifying activity. SPI is composed of β-conglycinin (7S) and conglycinin (11S), where 7S exhibited better emulsifying activity than 11S and played a significant role in the emulsifying activity of SPI [11]. The 7S consists of three subunits: α′~70 kDa, α~67 kDa, and β~50 kDa. The 7S-polyphenol nanoparticles or 7S also can be used as stabilizers to fabricate the outstanding antioxidant HIPE gels to be remarkably stable against heating or form HIPE with gel-like network structures [9,12]. Moreover, 7S–dextran conjugates had better emulsifying performance, especially the enhanced emulsion stability at the isoelectric point, compared to native 7S [13,14]. However, there has been less investigation into how the glycation treatment affects the structural characteristics and improves the emulsification activity of 7S by affecting the interfacial adsorption behavior at the oil/water interface.

Thus, this work aimed to analyze the mechanism of glycosylated 7S to improve their emulsification activity for stabilizing HIPE. Firstly, we investigated the effects of glycation with D-galactose on the physicochemical properties and structural characteristics of 7S. Secondly, we explored the effects of glycation on the interfacial properties of 7S at the oil/water interface, including interfacial adsorption behavior and dilatational rheology in the linear or non-linear region. Thirdly, we characterized the emulsifying performance of glycated 7S to stabilize the HIPE and tried to explain its emulsification mechanism. These findings are vital for understanding the relationships between interfacial adsorption and emulsifying properties and can be helpful for the utilization of protein–polysaccharide conjugates as emulsifiers in foods.

## 2. Materials and Methods

### 2.1. Materials

Soy β-conglycinin (85.0 ± 0.8%) was purified from defatted soybean powder (Shandong Yuwang Industrial Co., Ltd., China), according to our previous work [15]. Other reagents, including D-galactose (Gal), dodecane (purity > 99%), Nile red, and fluorescein isothiocyanate isomer I (FITC), were purchased from YuanYe Biotechnology Co., Ltd. (Shanghai, China).

### 2.2. Preparation of Glycated 7S

The 7S protein powder and D-galactose (Gal) were dispersed in 10 mmol/L phosphate buffer (pH 7.0 ± 0.2) under 2 h stirring conditions at room temperature. The sodium azide (2 mg/mL) was used as an antimicrobial agent. The pH value of 7S dispersion was adjusted to pH 7.0 ± 0.5 (2 mol/L NaOH), and then the insoluble proteins in 7S dispersion were removed (6000× *g*, 15 min) to obtain the 7S solution.

The equal volumes of 7S and Gal solutions (10 mg/mL) were mixed, and then the pH value of this mixture solution was adjusted to 7.0 ± 0.5. This 7S and Gal mixture solution was divided into three groups and then heated in a water bath at 70 °C for 12 h, 24 h, and 48 h, respectively. After that, the heated solutions were immediately cooled in an ice bath. The unreacted Gal was removed by dialysis tubes with a cut-off molecular weight of 7–14 kDa through dialyzing. Then the solutions were lyophilized to acquire the glycated 7S with different glycation times. The untreated 7S (native 7S) was used as the control.

### 2.3. Characterization of Untreated and Glycated 7S

#### 2.3.1. Measurement of Glycation Degree

The glycation extent was measured by the o-phthaldialdehyde (OPA) assay [16]. The samples (200 μL, 1 mg/mL) were mixed with OPA reagents (4 mL) and incubated in a water bath (35 °C) for 2 min. The absorbance of mixtures was measured at 340 nm, and the OPA reagent was regarded as the blank. The glycation degree was calculated according to Equation (1):(1)$$\;\mathrm{graft}\;\mathrm{degree}=\frac{A_{0}-A_{g}}{A_{0}}\;\times \;100\%$$
where A_0_ and A_g_ were the absorbances of untreated and glycated 7S, respectively.

#### 2.3.2. Hydrodynamic Diameter and Zeta Potential

Hydrodynamic diameter and zeta potentials of all samples were determined using a Zetasizer Nano-ZS instrument (Malvern Instruments Ltd., Worcestershire, UK) at 25 °C, according to our previous work [17]. All samples were diluted to avoid multiple scattering effects and were measured three times.

#### 2.3.3. Surface Hydrophobicity (H_0_)

The surface hydrophobicity (H_0_) of all samples was measured using 1-anilino-8- naphthalenesulfonate (ANS) (Sigma Aldrich, St. Louis, MO, USA) as the fluorescence probe. ANS solution was prepared in a phosphate buffer (0.01 mol/L, pH 7) at an 8 mmol/L concentration. The concentration of each protein sample was diluted to 0.05, 0.1, 0.2, 0.3, 0.4, and 0.5 mg/mL, respectively, with the same phosphate buffer. The ANS solution (20 μL) was added to each dilution (4 mL), and the fluorescence intensity (FI) was measured using an F4500 fluorescence spectrophotometer (Hitachi Co., Tokyo, Japan) with an excitation at 365 nm and emission at 484 nm. The slit width and PMT voltage were set at 5 nm and 400 V, respectively. The initial slopes of the fluorescence intensity versus protein concentration curves were used as H_0_.

#### 2.3.4. Intrinsic Fluorescence Spectroscopy

The intrinsic fluorescence spectra of all samples (1 mg/mL) were determined using an F4500 fluorescence spectrophotometer (Hitachi Co., Japan) with an excitation wavelength of 280 nm and an emission wavelength ranging from 300 nm to 450 nm. The slit, PMT voltage, and scan speed were set at 5 nm, 400 v, and 1200 nm/min, respectively.

#### 2.3.5. Far-UV Circular Dichroism Spectroscopy

Samples (0.1 mg/mL) were prepared according to the method in Section 2.2. The secondary structure of all protein samples was detected by far-UV circular dichroism spectrometry (JASCO; Tokyo, Japan) and was analyzed by SpectraManager software from JASCO Company. Distilled water was regarded as a blank. The scanning wavelength was set from 190 to 260 nm, and the speed was set at 100 nm/min.

#### 2.3.6. Interfacial Adsorption Kinetics Analysis

Samples (10 mg/mL) were prepared according to the method in Section 2.2. The interfacial tension of all samples at the oil/water interface was analyzed using an automatic drop tensiometer (Tracker Teclis/IT Concept, Longessaigne, France). The surface pressure (π) was calculated by Equation (2):(2)$$\pi =\gamma \;-\gamma_{0}\;$$
where γ is the surface tension of the sample solutions, and γ_0_ is the surface tension of the aqueous solution.

Moreover, diffusion rate (k_dif_), penetration rate (k_pen_), and rearrangement rate (k_rea_) were calculated according to our work published before [18].

#### 2.3.7. Dilatational Viscoelasticity Analysis

All samples (10 mg/mL) were prepared according to the method in Section 2.2. The dilatational viscoelasticity modulus (at a linear or non-linear region) of untreated and glycated 7S was measured by an automatic drop tensiometer. An equilibrium condition of the interface was obtained after adsorption of untreated and glycated 7S for 180 min. Then, the dilatational viscoelasticity modulus was measured as a function of time at 0.1 Hz and a 10% deformation amplitude (ΔA/A_0_ = 0.1) (in a linear region) or different deformations (ΔA/A_0_ = 0.05–0.30) (in the non-linear region). Lissajous plots of all samples were analyzed and calculated according to a method previously published [19].

### 2.4. Preparation for HIPE

The oil/water HIPE was fabricated by a one-pot homogenization process with an oil phase volume fraction of 0.8. The 4 mL of untreated and glycated 7S solutions was added to 16 mL of dodecane, and then the mixtures were sheared at 12,000 rpm for 2 min using an Ultra-Turrax T25 device (IKA-Werke GMBH & CO., Staufen, Germany).

### 2.5. Characterization of HIPE

#### 2.5.1. Droplet Size and Zeta Potential

The droplet size of freshly prepared HIPE was measured by Mastersizer 2000 (Malvern Instrument Co., Ltd., UK), in which the refractive index of dodecane and phosphate buffer solution were 1.560 and 1.330, respectively. Droplet size was reported as the volume-average diameter (d_(4.3)_).

The HIPE was diluted 100 times, and the Zeta-potential of HIPE was determined using ZS Zetasizer Nano (Malvern Instrument Co., Ltd., Malvern, UK) at 25 °C.

#### 2.5.2. Rheological Behavior Analysis

The rheological properties of HIPE stabilized by untreated and glycated 7S were measured using a strain-controlled rheometer (AR2000ex, TA, Crawley, USA) with a 40 mm parallel plate and a 1.0 mm test gap at 25 °C. The strain sweep test was performed by increasing the strain from 0.1% to 10% at a fixed frequency of 1.0 Hz. The frequency sweep test was performed by increasing the frequency from 0.1 Hz to 10 Hz at a fixed strain of 0.5% (a linear viscoelastic region).

#### 2.5.3. Stability of HIPE

The stability of HIPE stabilized by untreated and glycated 7S was measured by multiple light scattering using a Turbiscan Tower (Formulation Co., Toulouse, France). The freshly prepared HIPE (20 mL) was placed into the test tube. The backscattered and transmitted light of all samples was detected for 48 h.

### 2.6. Statistic

Graph plotting was performed by the software Origin 9.0 (OriginLab Co., Northampton, MA, USA). The data process was performed by an Excel 2016 (Microsoft Co., Washington, USA), and the software SPSS 19.0 (SPSS Inc., Chicago, IL, USA), and SPSS 19.0 were used to analyze significant differences between values (Duncan test).

## 3. Results and Discussion

### 3.1. Glycation Degree

As shown in Figure 1, the glycation degree (GD%) of glycated 7S significantly increased as the incubation time increased from 12 h to 48 h (*p* < 0.05). The ε-amino groups of 7S covalently bonded with sugar moieties promoted the glycation reactions, and, therefore, the GD% was significantly increased [20]. A similar glycation degree dependence on glycation time had been reported for glycinin glycated with soluble soybean polysaccharide [16], glycinin glycated with κ-carrageenan [1], and soy protein isolate glycated with glucose [21].

### 3.2. Particle Sizes and Zeta Potentials

The particle sizes of 7S were increased from 93 nm to 179 nm, and absolute zeta potentials were also increased from 19 mV to 32 mV, as the glycation degree increased (*p* < 0.05) (Figure 2A). The particle size of untreated 7S showed a unimodal distribution, while the particle size of glycated 7S showed a multimodal distribution, and the prominent peak of particle size shifted toward larger sizes (Figure 2B). All these results indicated that glycation treatment increased the particle sizes and net charges of 7S. This was because glycosylation led to the subunit dissociation of 7S, and the dissociated subunits recombined to form large 7S particles and exposed the charged groups to the surface of 7S. Similarly, soy protein isolates glycated with maltose resulted in protein aggregation with larger particle sizes and higher zeta potentials [22]. Peng et al. (2018) indicated that the introduction of negatively charged soy-soluble polysaccharides led to the partial subunit dissociation of glycinin [16]. The dissociated subunits interacted with each other to form new structures of glycinin with large sizes. Li et al. (2019) confirmed the generation of macromolecular polymer among soy protein isolate—glucose conjugates by analyzing the SDS-PAGE [21].

We questioned whether glycation treatment changed the protein’s structural characteristics or increased the particle sizes and net charges beneficial for oil/water interfacial adsorption. These questions will continue to be investigated by characterizing the structural features and the interfacial adsorption behavior of glycated 7S.

### 3.3. Intrinsic Fluorescence Spectra

The protein tertiary structure can be investigated by characterizing the changes in polarity around tryptophan residues using intrinsic fluorescence spectra. The fluorescence intensity of glycated 7S was sharply reduced with different glycation times, and the maximum absorption wavelength (λ_max_) was shown a noticeably red shift compared with untreated 7S (Figure 3A). The decreases in fluorescence intensity of glycated 7S could be related to the shielding effect of the saccharide chain around tryptophan residues. Meanwhile, the glycation led to the structural unfolding and increased the hydrophobic microenvironment of 7S, resulting in the red shift of the λ_max_ [23]. These results indicated glycated 7S had fewer compact tertiary structures than native 7S. Previous reports also indicated glycation changed the tertiary conformational state of glycated proteins such as whey protein isolates, soy protein isolates, and pea protein isolates [9,24,25].

### 3.4. Hydrophobicity

The fluorescence probe (ANS) can bind with hydrophobic groups on the surfaces of proteins. The glycation treatment exposed the hydrophobic residues and therefore increased the hydrophobicity (H_0_) of glycated 7S (*p* < 0.05), as shown in Figure 3B. This result was in line with 7S–dextran conjugates [14], soybean protein–dextran conjugates [26], and ultrasonically treated SPI—glucose conjugates [27]. Moreover, H_0_ of 24 h-glycated 7S and 48 h-glycated 7S was slightly decreased compared with 12 h-glycated 7S. This was because hydrophilic sugar moieties were introduced onto the surface of the 7S molecule owing to glycation, disrupting the hydrophilic/hydrophobic balance of 7S itself. Furthermore, thermal aggregation during glycation treatment led to re-embedding the originally exposed hydrophobic residues into the interior of the proteins. The above reasons led to a slight decrease in H_0_ [28]. It is worth noting that a relatively high H_0_ may improve an emulsifying capability since it can increase adsorption at an oil/water interface.

### 3.5. Far-UV Circular Dichroism Spectra

As shown in Table 1, the glycation treatment decreased the content of α-helix and β-turn but increased the content of β-sheet and the random coil of 7S. It indicated that introducing carbohydrate moieties into protein polypeptide chains changed the secondary structure of proteins. According to the report, the hydrogen bond was closely related to the stability of the α-helix region. Glycation treatment destructed the hydrogen bond and decreased α-helix content [21]. Moreover, a decline in α-helix content and an increase in β-sheet content indicated that the secondary structure of glycated 7S changed from an ordered one to a disordered one, increasing the molecular flexibility of the proteins. Additionally, Yu et al. (2018) found an irregular secondary structure in soy protein isolate hydrolysate–dextran conjugate. Similarly, soy protein isolate glycation with maltose [22] and β-conglycinin glycation with dextran led to a disordered secondary structure [29]. Investigations indicated that the flexible structure of proteins might benefit from interfacial adsorption and improved functional properties.

### 3.6. Interfacial Adsorption Behavior

The interfacial adsorption behavior of proteins was determined by analyzing the interfacial pressure (π) and adsorption kinetics of untreated and glycated 7S at the oil/water interface. As shown in Figure 4, both the untreated 7S and glycated 7S could be fast adsorbed onto the interfaces and increased the interfacial pressure (π). Furthermore, the glycated (12 h, 24 h, 48 h) 7S showed larger π values than untreated 7S, indicating that the former could be more effective in stabilizing the interface. The higher zeta potentials and larger sizes between glycated 7S could be favorable for occupying a larger surface area and forming a highly ordered layer at the interfaces, improving interfacial adsorption [2].

As shown in Table 2, the diffusion rate (K_dif_) and rearrangement rate (K_rea_) significantly increased dependent upon the glycation time. It indicated that the glycation treatment resulted in a flexible 7S structure and facilitated their structural unfolding and rearrangement at the oil/water interface. Similarly, glycation with gellan gum-endowed whey protein isolate enabled the structural flexibility to be efficiently adsorbed and unfolded structures at the oil/water interface [30]. Glycosylation between coconut protein isolate (CPI) and pectin made the protein structure loose and increased interfacial adsorption and rearrangement rates [8]. It was expected that glycated 7S with superior interfacial adsorption might exhibit better emulsification performance.

### 3.7. Dilatational Rheology Properties

The dilatational viscoelasticity can be used to assess the interaction between interfacial proteins and the resistance of formed interfacial protein layers to external deformation. As shown in Figure 5, the interfacial viscoelasticity modulus (E) of all samples rapidly increased as a function of time. The E values of glycated 7S with various glycation times were noticeably higher than untreated 7S. As discussed before, compared with untreated 7S, the glycation reaction increased the hydrophobicity and changed the structural flexibility of glycated 7S, which enhanced the protein interface interactions and therefore formed the viscoelastic protein layers [31]. Similar phenomena have been reported for sugar beet pectin–coconut protein conjugates [32] and ovalbumin–burdock polysaccharide conjugates [33], where E values of conjugates were higher than that of the single protein. These results indicated that glycation enhanced the viscoelasticity of adsorption layers and endued glycated 7S with the ability to resist external deformation, which was beneficial to the emulsification stability [34].

### 3.8. Non-Linear Dilatational Rheology Properties

The non-linear dilatational rheology can be characterized by analyzing the Lissajous plots. Our previous work used the Lissajous plots to investigate the dilatational rheology of 7S/pectin complexes in the non-linear viscoelastic region at the oil/water interface [18].

In the Lissajous plots, a linear plot, a spherical plot, and an elliptical plot represent the pure elastic response, the viscous response, and the viscoelastic response of an interface, respectively, when this interface is under large amplitude deformation [19]. As shown in Figure 6, the interface stabilized by untreated 7S showed a primarily elastic response, proven by the narrow and almost linear Lissajous plots at amplitudes up to 30%. These narrow plots indicated that the adsorbed 7S underwent a lower loss tangent at the interface and exhibited a relatively solid-like structure during deformation. The same result was found for the whey protein isolate-stabilized interface at a low concentration (0.1 wt%) [26] and the gliadin nanoparticle-stabilized interface [35], while the response of 24 h-glycated 7S-stabilized interfaces exhibited a viscoelastic contribution since the Lissajous plots were elliptical and became increasingly asymmetric at amplitudes up to 30%.

Meanwhile, the S factors were calculated to further quantify the degree of nonlinearity and are shown in Figure 7. The interfaces of untreated and 24 h-glycated 7S showed strain softening (S < 0) during extension and strain hardening (S > 0) during compression, respectively, with a dependency upon amplitudes. The softening behavior was progressively increased with increased amplitudes during extension, while the hardening behavior was decreased gradually with an increase in amplitudes during compression. The same change trends were also found for the whey protein-stabilized interface under amplitude up to 30% and 50% [26]. The pronounced strain softening behavior for glycated 7S-stabilized interfaces, particularly at amplitudes up to 30%, indicated that the interfacial layers were like viscoelastic liquid and started to flow in response to the large deformation. Thus, glycated 7S-stabilized interfaces were more stretchable than untreated 7S. These differences in the interfacial layer’s response to large deformation between untreated and glycated 7S may imply their capability to stabilize the emulsion under dynamic conditions.

On the other hand, for 24 h-glycated 7S-stabilized interfaces, the S factors of the Lissajous plots were always larger than untreated 7S in extension and compression, which indicated that the interfacial adsorption layers of the former had a more remarkable change in response to external deformation [26]. The 24-glycated 7S formed the viscoelastic adsorption layers and thus showed “fluidity” when dealing with deformation [36].

Based on the discussion above, it was reasonably inferred that glycated 7S formed a viscoelastic network at the oil/water interfaces due to structural flexibility and resisted deformation from large amplitudes [13]. However, the relatively rigid structure of untreated 7S did not facilitate the large deformation response at the interfacial. Thus, the Lissajous plots did not significantly change with the increased amplitudes. Similar results were found in soy glycinin–stevioside [37] and soy glycinin fibril-peptide system–steviol glycoside mixtures [38], as well as gliadin nanoparticles and gliadin–carboxymethyl cellulose nanoparticles [35].

### 3.9. Droplet Sizes and Zeta Potentials of HIPE

As shown in Figure 8A, droplet sizes of glycated 7S-stabilized HIPE were significantly decreased compared to untreated 7S. The zeta potentials of the former were noticeably increased compared to those of the latter. Moreover, as shown in Figure 8B, the droplet size of all HIPE showed a unimodal distribution. The increased emulsifying performance of 7S may be related to the following two reasons. The increased structural flexibility due to glycation facilitated structural rearrangement and formed the viscoelastic layers of 7S at the interface, thereby effectively improving the emulsifying performance. According to the reports, the emulsifying activity of ultrasonication-treated soy protein isolate–glucose conjugates was positively related to their structural changes [21,27] (Cui et al., 2020; Li et al., 2019). The emulsion activity index of coconut protein isolate (CPI)–pectin conjugates were significantly higher than that of CPI–pectin mixtures or CPI, which was closely related to their structural flexibility [8]. Furthermore, the saccharides could provide steric repulsions between adsorbed layers to inhibit the aggregation of oil droplets [32] (Zhou et al., 2021). For example, Li et al. (2019) found that the steric hindrance provided by soy protein isolate–glucose conjugates inhibited the oil droplet aggregation, thereby reducing the particle sizes of emulsions [21].

### 3.10. Rheological Analysis of HIPE

The effectiveness of glycated 7S to stabilize the HIPE may be related to the formation of the gel network and could be characterized by rheological analysis. As shown in Figure 9A, the elastic moduli (*G′*) and viscous moduli (*G″*) of HIPE stabilized by the untreated and glycated 7S progressively increased with the increased frequency, indicating the formation of a densely packed interfacial layer. Moreover, the *G′* and *G″* of HIPE stabilized by glycated 7S were always larger than those of untreated 7S, as shown in Figure 9B. This was because glycated 7S formed a gel-like network in the HIPE with strengthened stiffness [5]. Sun et al. (2022) also found that glycated pea protein isolate can form a gel network in HIPE and enhance their stiffness and viscoelasticity. Similar findings had been reported for HIPE stabilized by soy glycinin–soy polysaccharide conjugates [9] and soy protein isolate–D-galactoses conjugates [25].

### 3.11. Stability of HIPE

To test the stability of HIPE, Turbiscan Tower was used to measure the backscattered light of all HIPE in the 48 h scanning process to calculate the emulsion stability index (TSI index). As shown in Figure 10, the TSI index of HIPE stabilized by untreated 7S was progressively increased, while there was no significant change for HIPE stabilized by glycated 7S. It was verified that the glycated proteins could be closely packed to form a thick adsorption layer at the interfaces, providing strong steric repulsions and improving the emulsion stability [39]. Feng et al. (2021) also found that compact viscoelastic interfacial layer and gel-like network structure endued HIPEs with good long-term storage stability [40]. In addition, the formation of bridging flocculation between two oil droplets in HIPE might contribute to their stability. This mechanism was also confirmed in the HIPE stabilized by ovalbumin [41] and β-conglycinin [42]. Thus, the stability of HIPE with self-supportive structures prepared by glycated 7S may be primarily associated with steric repulsions due to the thick interfacial adsorption layers and the formation of bridge emulsions.

## 4. Conclusions

The authors speculated the effects of glycation on the interfacial adsorption and emulsifying performance of glycated 7S based on the results mentioned above. The glycation treatment led to the dissociation and recombination of the 7S subunit, increasing particle sizes and zeta potentials of glycated 7S. The higher zeta potentials and larger sizes between glycated 7S could be favorable for occupying a larger surface area and forming a highly ordered layer at the interfaces, improving interfacial adsorption. As the glycation reaction increased, the tertiary structures of proteins were destroyed and unfolded, exposing the hydrophobic groups buried within the proteins. Each secondary structure gradually transformed from an ordered one to a disordered one. These results caused glycated 7S to be more flexible than untreated 7S, which was beneficial for the structural unfolding and arrangement of glycated 7S, improving interfacial adsorption and forming compact interfacial viscoelasticity layers at the oil/water interface. As a result, the increased interfacial adsorption behavior contributed to the improved emulsifying performance of glycated 7S to stabilize HIPE with self-supportive structures. Furthermore, elastic gel-like networks of HIPE stabilized by glycated 7S led to emulsification stability. This work could provide new insight into the relationships between interfacial and emulsifying properties and could also be useful for the utilization of protein–polysaccharide conjugates as emulsifiers in foods.

## Figures and Tables

**Figure 1 foods-12-02706-f001:**
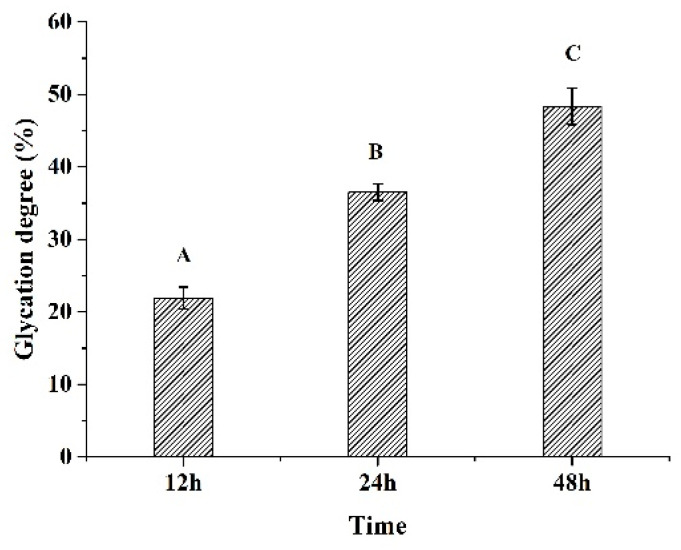
The effects of glycation time on the glycation degree. (Different letters indicated a significant difference).

**Figure 2 foods-12-02706-f002:**
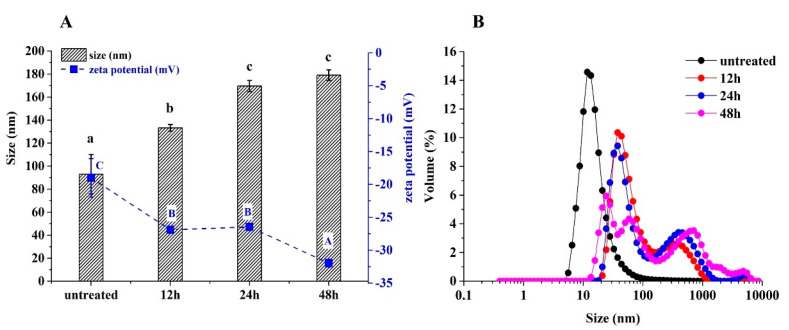
The particle sizes and zeta potentials (**A**) and size distributions (**B**) of untreated and glycated 7S with different glycation times. (Different letters with the same colors in (**A**) indicate a significant difference).

**Figure 3 foods-12-02706-f003:**
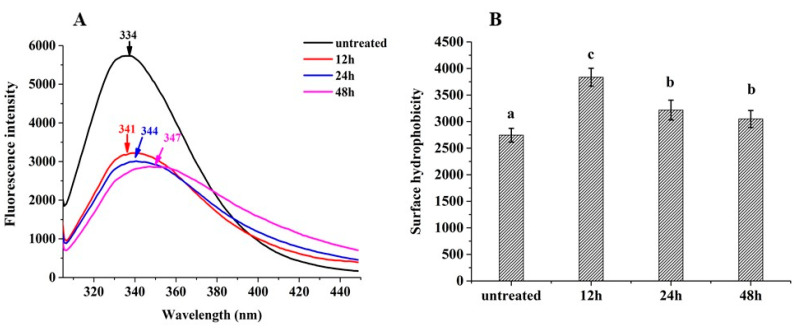
The fluorescence spectra (**A**) and surface hydrophobicity (**B**) of untreated and glycated 7S with different glycation times. (Different letters in (**B**) indicate a significant difference).

**Figure 4 foods-12-02706-f004:**
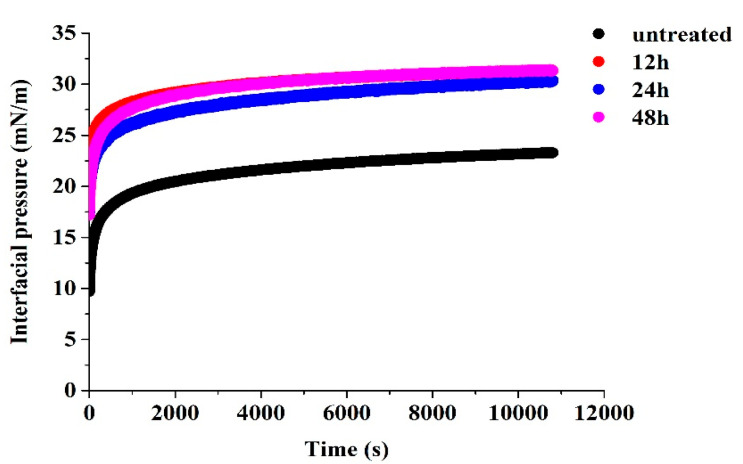
The interfacial pressure of untreated and glycated 7S with different glycation times at the oil/water interface as a function of time.

**Figure 5 foods-12-02706-f005:**
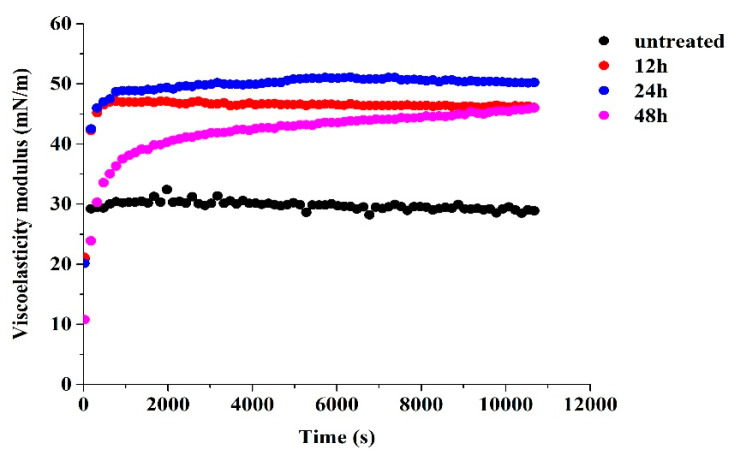
The interfacial viscoelasticity modulus of untreated and glycated 7S with different glycation times at the oil/water interface as a function of time.

**Figure 6 foods-12-02706-f006:**
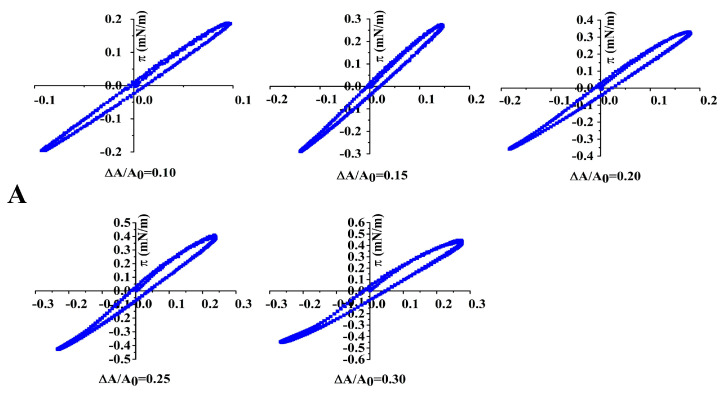
The Lissajous plots of untreated (**A**) and 24 h-glycated 7S (**B**) at the oil/water interface at various amplitudes of deformation (ΔA/A_0_ = 0.05–0.30).

**Figure 7 foods-12-02706-f007:**
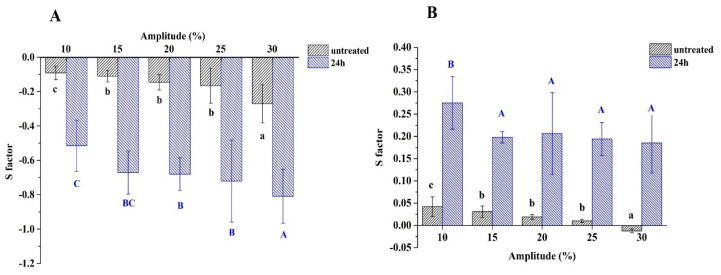
The S factors of untreated and 24 h-glycated 7S during extension (**A**) and compression (**B**) under various amplitudes of deformation (10–30%). (Different letters with the same colors indicate a significant difference).

**Figure 8 foods-12-02706-f008:**
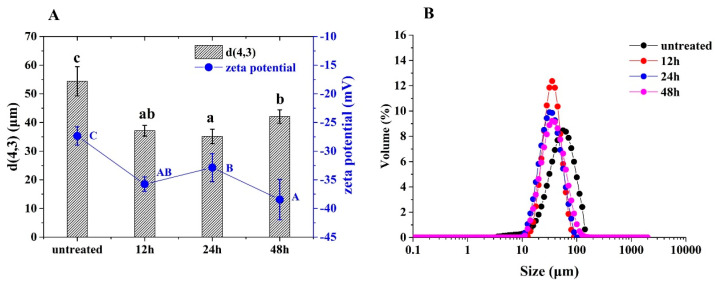
The droplet sizes and zeta potentials (**A**) and size distributions (**B**) of HIPE stabilized by untreated and glycated 7S with different glycation times. (Different letters with the same colors in (**A**) indicate a significant difference).

**Figure 9 foods-12-02706-f009:**
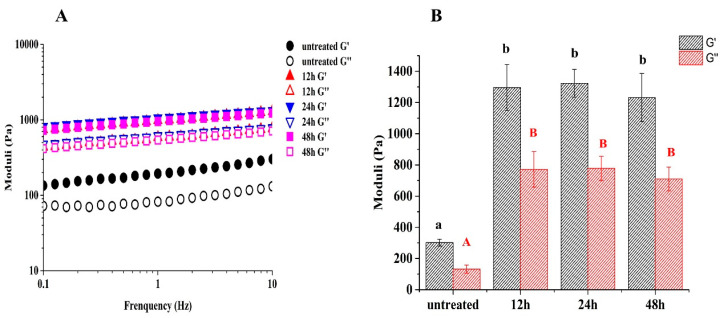
The profiles of total moduli against increasing frequency (0.1–10 Hz) (**A**), as well as elastic (*G′*) and viscous (*G″*) moduli (strain 0.5%, frequency 10 Hz) **(B**) for the HIPE stabilized by untreated and glycated 7S with different glycation times. (Different letters with the same colors in (**B**) indicate a significant difference).

**Figure 10 foods-12-02706-f010:**
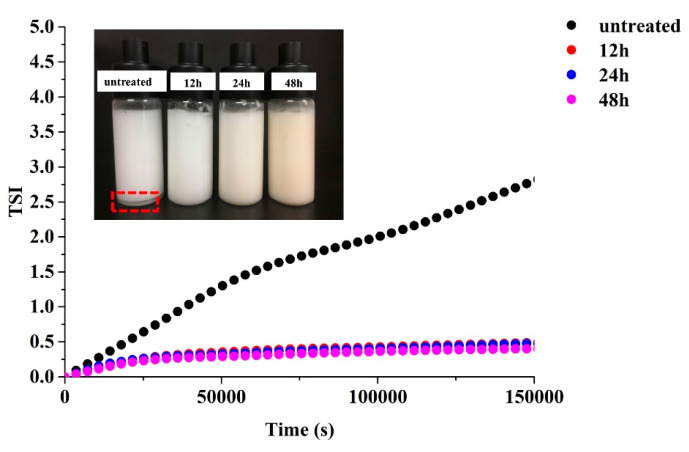
The stability of high internal phase emulsions stabilized by untreated and glycated 7S with different glycation times.

**Table 1 foods-12-02706-t001:** Secondary structure composition (%) of untreated and glycated 7S with different glycation times.

	α-Helix	β-Sheet	β-Turn	Random Coil
untreated	18.3 ± 0.9 b	37.3 ± 2.5 a	14.2 ± 2.2 b	30.2 ± 1.6 a
12 h	16.4 ± 0.9 b	40.6 ± 2.0 ab	11.4 ± 2.7 a	31.6 ± 1.4 a
24 h	12.3 ± 1.1 a	41.2 ± 0.9 ab	10.5 ± 2.1 a	36.0 ± 1.6 b
48 h	11.7 ± 2.7 a	43.3 ± 4.0 b	10.3 ± 1.9 a	34.6 ± 2.2 b

Note: Different letters in the same column indicate a significant difference.

**Table 2 foods-12-02706-t002:** Adsorption kinetics of untreated and glycated 7S with different glycation times at the oil/water interface.

	K_dif_ (mN/m s^0.5^)	K_pen_ × 10^−4^ (s^−1^)	K_rea_ × 10^−4^ (s^−1^)
untreated	0.39 ± 0.02 a	−2.85 ± 0.00 b	−17.49 ± 0.00 b
12 h	0.49 ± 0.06 b	−3.29 ± 0.00 a	−21.77 ± 0.00 b
24 h	0.55 ± 0.03 bc	−2.76 ± 0.00 b	−34.44 ± 0.00 a
48 h	0.64 ± 0.08 c	−3.69 ± 0.00 a	−39.54 ± 0.00 a

Note: K_dif_: diffusion rate; K_pen_: penetration rate; K_rea_: rearrangement rate. Different letters in the same column indicate a significant difference.

## Data Availability

The data used to support the findings of this study can be made available by the corresponding author upon request.

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
