# Peer review of "Glycation Improved the Interfacial Adsorption and Emulsifying Performance of β-Conglycinin to Stabilize the High Internal Phase Emulsions"

_foods, 2023, doi:10.3390/foods12142706_

Round 1

Reviewer 1 Report

This interesting study investigates the effect of glycation on interfacial adsorption and emulsifying performance of β-conglycinin to stabilize the high internal phase emulsions. The manuscript was well structured. However, a minor revision is needed. Following, are my comments. 

Please improve the resolution of Figure 8 and enlarge all fonts.

2.2. The concentrations of 7S and Gal solutions were not mentioned.

2.5.3. It was mentioned that the scanning of turbidity was performed during 24 h; however, the scanning was shown up to 48 h in Figure 10.

3.10. Both G′ and G″ should be written in italic form.

-The space between numbers and their relevant units should be considered in the whole text (such as 24 h, 70 ℃, etc.)

Figure 10: Why the HIPE sample tended to yellowness after exposure to longer glycation time?

-Please revise the references. In some of them, the abbreviation of the journal title was written.

Reviewer 2 Report

* More detail regarding the hypothesis of the manuscript could be given. The application could be novel but why is necessary to apply such systems for these materials? 

*Materials and method parts were written- and organized-well.

* Results and discussion part should be improved. In general, the authors compared their results with previous studies. But it is necessary to explain possible reasons for the results. Also, the results found in previous studies could be presented in detail instead of similar results..

*Conclusion part should be improved. Why were the findings important for scientific literature and industry? etc.

*Minor editing of English language required.

Reviewer 3 Report

Section 2.1- powder is spelled wrong.

What is the remaining 15% of the soy B-conglycinin? Other proteins, carbohydrates?

Section 2.2- Were the 7S and D-galactose dispersed in separate buffered solutions and then mixed together after the 7S was centrifuged? How much of the 7S was removed by centrifugation? What were the concentrations of these solutions?

Page 3, first line- do you mean removed?

Sections 2.3.5-2.3.7- Preparations of the samples for analysis were not described.

conclusions- The mechanism of the emulsification mechanism may have been explained, but this may not be helpful in using protein-polysaccharide conjugates as emulsifiers in food. Experiments with real food comparing the conjugates to un-modified soy protein isolates, or whatever is used commercially as emulsifiers, would be a more convincing conclusion.

The paper would be easier to read if the English grammar and spelling was improved. Please have a native English-speaking person edit the paper.

Round 2

Reviewer 3 Report

Line 178- How can the test gap in the rheometer be 1.0 nm when the particle size of the material is more than 93 nm?
